# Safe Nonlinear Control Using Robust Neural Lyapunov-Barrier Functions

**Charles Dawson**[1], **Zengyi Qin**[1], **Sicun Gao**[2], **Chuchu Fan**[1]
[1] Massachusetts Institute of Technology, {`cbd, qinzy, chuchu`}`@mit.edu`
[2] University of California, San Diego, `sicung@ucsd.edu`

**Keywords:** Certified control, learning for control

**Abstract:** Safety and stability are common requirements for robotic control systems; however, designing safe, stable controllers remains difficult for nonlinear and uncertain models. We develop a model-based learning approach to synthesize robust feedback controllers with safety and stability guarantees. We take inspiration from robust convex optimization and Lyapunov theory to define robust control Lyapunov barrier functions that generalize despite model uncertainty. We demonstrate our approach in simulation on problems including car trajectory tracking, nonlinear control with obstacle avoidance, satellite rendezvous with safety constraints, and flight control with a learned ground effect model. Simulation results show that our approach yields controllers that match or exceed the capabilities of robust MPC while reducing computational costs by an order of magnitude. We provide source code at `github.com/dawsonc/neural_clbf/`.

## 1 Introduction

Robot control systems are challenging to design, not least because of the problems of *task complexity* and *model uncertainty*. Robotics control problems like those in Fig. 1 often involve both safety and stability requirements, where the controller must drive the system towards a goal state while avoiding unsafe regions. Complicating matters, the model used to design the controller is seldom a perfect representation of the physical plant, and so controllers must account for uncertainty in any parameters (e.g. mass, friction, or unmodeled effects) that vary between the engineering model and true plant. Automatically synthesizing safe, stable, and robust controllers for

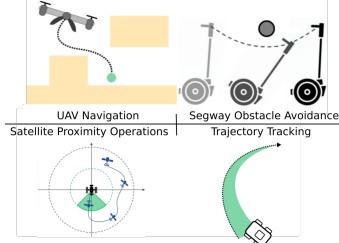

Figure 1: Safe control problems considered in Section 6.

nonlinear reach-avoid tasks is a long-standing open problem in controls. In this paper, we address this problem with a novel approach to robust model-based learning. Our work presents a unified framework for handling both model uncertainty and complex safety and stability specifications.

Over the years, several approaches have been proposed to solve this problem. In one view, reach-avoid can be treated as an optimal control problem and solved using model predictive control (MPC) schemes and their robust variants. Robust MPC promises a method for general-purpose controller synthesis, finding an optimal control signal given only a model of the system and a specification of the task. However, there are a number of recognized disadvantages of robust MPC. First, there are currently no techniques for guaranteeing the safety, stability, or recursive feasibility of robust MPC beyond the linear case [1]. Second, model uncertainty (e.g. mass or friction) is often multiplicative in the dynamics, but robust MPC is typically limited to additive uncertainty [1, 2]. Finally, MPC is computationally expensive, making it difficult to achieve high control frequencies in practice [3].

An alternative method for synthesizing safe, stable controllers comes from Lyapunov theory, through the use of control Lyapunov and control barrier functions (resp., CLFs and CBFs, [4]) — certificates that prove the stability and safety of a control system, respectively. CLFs and CBFs are similar to standard Lyapunov and barrier functions, but they can be used to synthesize a controller rather than

5th Conference on Robot Learning (CoRL 2021), London, UK.

just verifying the performance of a closed-loop system. Unfortunately, CLF and CBF certificates are very difficult construct in general, particularly for systems with nonlinear dynamics [5].

The most recent set of methods promising general-purpose controller synthesis come from the field of learning for control; for instance, using reinforcement learning [6, 7] or supervised learning [8, 9, 10, 11]. However, the introduction of learning-enabled components into safety-critical control tasks raises questions about soundness, robustness, and generalization. Some learning-based control techniques incorporate certificates such as Lyapunov functions [8], barrier functions [12, 10, 13], and contraction metrics [9, 11] to prove the soundness of learned controllers. Unfortunately, these certificates' guarantees are sensitive to uncertainties in the underlying model. In particular, if the model used during training differs from that encountered during deployment, then guarantees on safety and stability may no longer hold.

Our main contribution is a learning-based framework for synthesizing robust nonlinear feedback controllers from safety and stability specifications. This contribution has two parts. First, we provide a novel extension of control Lyapunov barrier functions to robust control, defining a robust control Lyapunov barrier function (robust CLBF). Second, we develop a model-based approach to learning robust CLBFs, which we use to derive a safe controller using techniques from robust convex optimization. Other methods for learning Lyapunov and barrier certificates exist, but a key advantage of our approach is that we learn certificates with explicit robustness guarantees, enabling generalization beyond the system parameters seen during training. We demonstrate our approach on a range of challenging control problems, including trajectory tracking, nonlinear control with obstacle avoidance, flight control with a learned model of ground effect, and a satellite rendezvous problem with non-convex safety constraints, comparing our approach with robust MPC. In all of these experiments, we find that our method either matches or exceeds the performance of robust MPC while reducing computational cost at runtime by at least a factor of 10. We provide source code at github.com/dawsonc/neural_clbf/ and video at youtu.be/4MWVLtURxG0.

## 2 Related Work

This work builds on a rich history of certificate-based control theory, including classical Lyapunov functions as well as more recent approaches such as control Lyapunov functions (CLFs [14, 15]) and control barrier functions (CBFs [16], a generalization of artificial potential fields [17]). The majority of classical certificate-based controllers rely on hand-designed certificates [18, 19], but these can be difficult to obtain for nonlinear or high-dimensional systems. Some automated techniques exist for synthesizing CLFs and CBFs; however, many of these techniques (such as finding a Lyapunov function as the solution of a partial differential equation) are computationally intractable for many practical applications [5]. Other automated synthesis techniques are based on convex optimization, particularly sum-of-squares programming (SOS, [20]), but are limited to systems with polynomial dynamics and do not scale favorably with the dimension of the system.

A promising line of work in this area is to use neural networks to learn certificate functions. These techniques range in complexity from verifying the stability of a given control system [21, 22] to simultaneously learning a control policy and certificate [9, 8, 10]. Most of these works do not explicitly consider robustness to model uncertainty, although contraction metrics may be used to certify robustness to bounded additive disturbance [9].

Most approaches to handling model uncertainty in the context of certificate-guided learning for control involve online adaptation. For example, [18, 23] assume that a CLF or CBF are given and learn the unmodeled residuals in the CLF and CBF derivatives. When combined with a QP-based CLF/CBF controller, this technique enables adaptation to model uncertainty but relies on a potentially unsafe exploration phase. Although safe adaptation strategies exist, the main drawback with these techniques is their reliance on a hand-designed CLF and CBF, which are non-trivial to synthesize for nonlinear systems. Additionally, combined CLF/CBF controllers are prone to getting stuck when the feasible sets of the CLF and CBF no longer intersect.

Online optimization-based control techniques such as model-predictive control (MPC) are also relevant as a general-purpose control synthesis strategy. However, the computational complexity of MPC, and particularly robust MPC, is a widely-recognized issue, particularly when considering deployment to resource-constrained robotic systems such as UAVs [1, 3]. We revisit the computational cost of robust MPC, particularly as compared with the cost of our proposed method, in Section 6.

Some approaches apply learning to characterize uncertainty in system dynamics and augment a robust MPC scheme [24], but these methods do not fundamentally change the computational burden of MPC. Other methods rely on imitation learning to recreate an MPC-based policy online [25], but these methods can encounter difficulties in generalizing beyond the training dataset.

A number of techniques from classical nonlinear control also deserve mention, such as sliding mode and adaptive controllers. These methods do not directly support state constraints and so must be paired with a separate trajectory planning layer [26]. Another drawback is that these techniques require significant effort to manually derive appropriate feedback control laws, and we are primarily interested in automated techniques for controller synthesis.

## 3 Preliminaries and Background

We consider continuous-time, control-affine dynamical systems of the form $\dot{x} = f_\theta(x) + g_\theta(x)u$, where $x \in \mathcal{X} \subseteq \mathbb{R}^n$, $u \in \mathbb{R}^\ell$, and $f_\theta : \mathbb{R}^n \to \mathbb{R}^n$ and $g_\theta : \mathbb{R}^n \to \mathbb{R}^{n \times \ell}$ are smooth functions modeling control-affine nonlinear dynamics. We assume that $f_\theta$ and $g_\theta$ depend on model parameters $\theta \in \Theta \subseteq \mathbb{R}^r$ and are affine in those parameters for any fixed $x$. This assumption on the dynamics is not restrictive; it covers many physical systems with uncertainty in inertia, damping, or friction (e.g. rigid-body dynamics or systems described by the manipulator equations), and it includes bounded additive and multiplicative disturbance as a special case. We also assume that $f_\theta$ and $g_\theta$ are Lipschitz but make no further assumptions, allowing us to consider cases when components of $f_\theta$ and $g_\theta$ are learned from experimental data. For concision, we will use $f$ and $g$ (without subscript) to refer to the dynamics evaluated with nominal parameters $\theta_0 \in \Theta$. In this paper, we consider the following control synthesis problem:

**Definition 1** (Robust Safe Control Problem). *Given a control-affine system with uncertain parameters $\theta \in \Theta$, a goal configuration $x_{\text{goal}}$, a set of unsafe states $\mathcal{X}_{\text{unsafe}} \subseteq \mathcal{X}$, and a set of safe states $\mathcal{X}_{\text{safe}} \subseteq \mathcal{X}$ (such that $\mathcal{X}_{\text{safe}} \cap \mathcal{X}_{\text{unsafe}} = \emptyset$ and $x_{\text{goal}} \in \mathcal{X}_{\text{safe}}$), find a control policy $u = \pi(x)$ such that all trajectories $x(t)$ satisfying $\dot{x} = f_\theta(x) + g_\theta(x)\pi(x)$ and $x(0) \in \mathcal{X}_{\text{safe}}$ have the following properties for any parameters $\theta$:*

| | |
|---|---|
| ***Reachability** of $x_{\text{goal}}$ with tolerance $\delta$: $\lim_{t\to\infty} \|x(t) - x_{\text{goal}}\| \leq \delta$* | ***Safety:** $x(t_1) \in \mathcal{X}_{\text{safe}}$ implies $x(t_2) \notin \mathcal{X}_{\text{unsafe}} \, \forall \, t_2 \geq t_1$* |

Simply put, we wish to *reach* the goal $x_{\text{goal}}$ while *avoiding* the unsafe states $\mathcal{X}_{\text{unsafe}}$. We use the notion of reachability instead of asymptotic stability to permit (small) steady-state error; in the following we will use "stable" as shorthand for reachability. Note that we do not require $\mathcal{X}_{\text{safe}} \cup \mathcal{X}_{\text{unsafe}} = \mathcal{X}$, as it will be made clear in the following discussion that we need a non-empty boundary layer $\mathcal{X} \setminus (\mathcal{X}_{\text{safe}} \cup \mathcal{X}_{\text{unsafe}})$ to allow for flexibility in finding a safety certificate.

Lyapunov theory provides tools that are naturally suited to reach-avoid problems: control Lyapunov functions (for stability) and control barrier functions (for safety [4]). To avoid issues arising from learning two separate certificates, we rely on a single, unifying certificate known as a control Lyapunov barrier function (CLBF). Our definition of CLBFs is related to those in [27] and [28] (differing from the formulation in [27] by a constant offset $c$, and differing from [28] where safety and reachability are proven using two separate CLBFs). We begin by providing a standard definition of a CLBF in the non-robust case, but in the next section we provide a novel, robust extension of CLBF theory before demonstrating how neural networks may be used to synthesize these functions for a general class of dynamical system. In the following, we denote $L_f V$ as the Lie derivative of $V$ along $f$.

**Definition 2** (CLBF). *A function $V : \mathcal{X} \to \mathbb{R}$ is a CLBF if, for some $c, \lambda > 0$,*

$$V(x_{\text{goal}}) = 0 \quad \text{(1a)} \qquad\qquad V(x) \leq c \, \forall \, x \in \mathcal{X}_{\text{safe}} \quad \text{(1c)}$$

$$V(x) > 0 \, \forall \, x \in \mathcal{X} \setminus x_{\text{goal}} \quad \text{(1b)} \qquad V(x) > c \, \forall \, x \in \mathcal{X}_{\text{unsafe}} \quad \text{(1d)}$$

$$\inf_u L_f V + L_g V u + \lambda V(x) \leq 0 \, \forall \, x \in \mathcal{X} \setminus x_{\text{goal}} \quad \text{(1e)}$$

Intuitively, we can think of a CLBF as a special case of a control Lyapunov function where the safe and unsafe regions are contained in sub- and super-level sets, respectively. If we define a set of admissible controls $K(x) = \{u \mid L_f V + L_g V u + \lambda V \leq 0\}$, then we arrive at a theorem proving the stability and safety of any controller that outputs elements of this set (the proof is included in the supplementary material).

**Theorem 1.** *If $V(x)$ is a CLBF then any control policy $\pi(x) \in K(x) \, \forall \, x \in \mathcal{X}$ will be both safe and stable, in the sense of Definition 1.*

Based on these results, we can define a CLBF-based controller, analogous to the CLF/CBF-based controller in [18] but without the risk of conflicts between the CLF and CBF conditions, relying on the CLBF $V$ and some nominal controller $\pi_{\text{nominal}}$ (e.g. the LQR policy):

$$\pi_{\text{CLBF}}(x) = \arg\min_u \quad \frac{1}{2}\|u - \pi_{\text{nominal}}(x)\|^2 \qquad \text{(CLBF-QP)}$$

$$\text{s.t.} \quad L_f V + L_g V u + \lambda V \leq 0 \qquad (2)$$

It should be clear that $\pi_{\text{CLBF}}(x) \in K(x) \ \forall \ x \in \mathcal{X} \setminus x_{\text{goal}}$, so this controller will result in a system that is certifiably safe and stable (with the CLBF $V$ acting as the certificate). The nominal control signal $\pi_{\text{nominal}}$ is included to encourage smoothness in the solution $\pi_{\text{CLBF}}(x)$, particularly near the desired fixed point at $x_{\text{goal}}$ where $\dot{V}$ becomes small. CLBFs provide a single, unified certificate of safety and stability; however, some significant issues remain. In particular, how do we guarantee that a CLBF will generalize beyond the nominal parameters?

## 4 Robust CLBF Certificates for Safe Control

In this section, we extend the definition of CLBFs to provide explicit robustness guarantees, and we present a key theorem proving the soundness of robust CLBF-based control.

**Definition 3** (Robust CLBF, rCLBF). *A function $V : \mathcal{X} \to \mathbb{R}$ is a robust CLBF for bounded parametric uncertainty $\theta \in \Theta$, where $\Theta$ is the convex hull of scenarios $\theta_1, \theta_2, \ldots, \theta_{n_s}$ if the standard CLBF conditions* (1a)–(1d) *hold, the dynamics $f$ and $g$ are affine with respect to $\theta$, and $\forall \ x \in \mathcal{X} \setminus x_{\text{goal}}$ there exist $c, \lambda > 0$ such that*

$$\inf_u L_{f_{\theta_i}} V + L_{g_{\theta_i}} V u + \lambda V(x) \leq 0 \qquad \forall i = 1, \ldots, n_s \qquad (3)$$

As in the non-robust case, we define the set of admissible controls for a robust CLBF, $K_r(x) = \left\{ u \mid L_{f_{\theta_i}} V + L_{g_{\theta_i}} V u + \lambda V \leq 0 \ \forall \ i = 0, \ldots, n_s \right\}$, and the corresponding QP-based controller, the soundness of which is given by Theorem 2:

$$\pi_{\text{rCLBF}} = \arg\min_u \|u - \pi_{\text{nominal}}\|^2 \qquad \text{(rCLBF-QP)}$$

$$\text{s.t.} \quad L_{f_{\theta_i}} V + L_{g_{\theta_i}} V u + \lambda V \leq 0; \ i = 0, \ldots, n_s \qquad (4)$$

**Theorem 2.** *If $V(x)$ is a robust CLBF, then any control policy $\pi(x) \in K_r(x) \ \forall \ x \in \mathcal{X}$ will be both safe and stable, in the sense of Definition 1, when executed on a system $f_\theta, g_\theta$ with uncertain parameters $\theta \in \Theta$ (where $\Theta$ is the convex hull of scenarios $\theta_0, \ldots, \theta_{n_s}$).*

*Proof.* See the supplementary materials. □

This result demonstrates the soundness and robustness of an rCLBF-based controller, but does not provide a means to construct a valid rCLBF. In the next section, we will present an automated model-based learning approach to rCLBF synthesis, yielding a general framework for solving robust safe control problems even for systems with complex, nonlinear, or partially-learned dynamics.

## 5 Learning Robust CLBFs

A persistent challenge in using of certificate-based controllers is the difficulty of finding valid certificates, especially for systems with nonlinear dynamics and complex specifications of $\mathcal{X}_{\text{safe}}$ and $\mathcal{X}_{\text{unsafe}}$ (e.g. obstacle avoidance). Taking inspiration from recent advances in certificate-guided learning for control [8, 10], we employ a model-based supervised learning framework to synthesize an rCLBF-based controller. The controller architecture is comprised of three main parts: the rCLBF $V$, a proof controller $\pi_{\text{NN}}$, and the QP-based controller (rCLBF-QP). We parameterize $V : \mathcal{X} \to \mathbb{R}$ and $\pi_{\text{NN}} : \mathcal{X} \to \mathbb{R}^\ell$ as neural networks. These networks are trained offline, where $\pi_{\text{NN}}$ is used to prove that the feasible set of (rCLBF-QP) is non-empty, then $V$ is evaluated online to provide the parameters of (rCLBF-QP), which is solved to find the control input. In the offline training stage, our primary goal is finding an rCLBF $V(x)$ such that the conditions of Definition 3 are satisfied. To ensure (1b), we define $V(x) = w^T(x)w(x) \geq 0$, where $w$ is the activation vector of the last

hidden layer of the $V$ neural network. To train $V$ such that conditions (1a), (1c), (1d), and (3) are satisfied over the domain of interest, we sample $N_{\text{train}}$ points uniformly at random from $\mathcal{X}$ to yield a population of training points $x$, then define the empirical loss:

$$\mathcal{L}_{\text{rCLBF}} = V(x_{\text{goal}})^2 + a_1 \frac{1}{N_{\text{safe}}} \sum_{x \in \mathcal{X}_{\text{safe}}} [\epsilon + V(x) - c]_+ + a_2 \frac{1}{N_{\text{unsafe}}} \sum_{x \in \mathcal{X}_{\text{unsafe}}} [\epsilon + c - V(x)]_+$$

$$+ \frac{a_3}{n_s N_{train}} \sum_x r(x) \sum_{i=0}^{n_s} [\epsilon + L_{f_{\theta_i}} V(x) + L_{g_{\theta_i}} V(x) \pi_{\text{NN}}(x) + \lambda V(x)]_+ \qquad (5)$$

where $a_1 - a_3$ are positive tuning parameters, $\epsilon > 0$ is a small parameter (typically 0.01) that allows us to encourage strict inequality satisfaction and enables generalization claims, $N_{\text{safe}}$ and $N_{\text{unsafe}}$ are the number of points in the training sample in $\mathcal{X}_{\text{safe}}$ and $\mathcal{X}_{\text{unsafe}}$, respectively, and $[\circ]_+ = \max(\circ, 0)$ is the ReLU function. The terms in this empirical loss are directly linked to conditions (1a), (1c), (1d), and (3) such that each term is zero if the corresponding condition is satisfied at all $N_{\text{train}}$ training points. For example, the final term in this loss is designed to encourage satisfaction of the robust CLBF decrease condition (3). The factor $r(x)$ in the final term is computed by solving (rCLBF-QP) at each training point and computing the maximum violation of constraint (4), such that $r(x) = 0$ when the QP has a feasible solution and $r(x) > 0$ otherwise. This loss is optimized using stochastic gradient descent, alternating epochs between training the $V$ and $\pi_{\text{NN}}$ networks. During training, we rely on $\pi_{\text{NN}}$ to compute the time derivative of $V(x)$ in the final term of the loss. To provide a training signal for $\pi_{\text{NN}}$, we define an additional loss $\mathcal{L}_\pi = \|\pi_{\text{NN}} - \pi_{\text{nominal}}\|^2$, where $\pi_{\text{nominal}}$ is a nominal controller (e.g. a policy derived from an LQR approximation). The parameters of $V$ and $\pi_{\text{NN}}$ are optimized using the combined loss $\mathcal{L} = \mathcal{L}_{\text{rCLBF}} + (10^{-5})\mathcal{L}_\pi$. The small weight applied to $\mathcal{L}_\pi$ ensures that the training process prioritizes satisfying the CLBF conditions.

An important detail of our control architecture is that the learned control policy $\pi_{\text{NN}}$ is used primarily to demonstrate that the feasible set of (rCLBF-QP) is non-empty. We are not required to use $\pi_{\text{NN}}$ at execution time; we can choose any control policy from the admissible set $K_r(x)$. In the online stage, we rely on an optimization-based controller (rCLBF-QP), which solves a small quadratic program with $n_s$ constraints and $\ell$ variables (one for each element of $u$). To ensure that this QP is feasible at execution, we permit a relaxation of the CLBF constraints (4) and penalize relaxation with a large coefficient in the objective. Once trained, $V$ can be verified using neural-network verification tools [29], sampling [30], or a generalization error bound [10]. More details on data collection, training, implementation, and verification strategies are included in the supplementary materials.

It is important to note that this training strategy encourages satisfying (3) only on the finite set of training points sampled uniformly from the state space; there is no learning mechanism that enforces dense satisfaction of (3). In the supplementary materials, we include plots of 2D sections of the state space showing that (3) is satisfied at the majority of points, but there is a relatively small violation on a sparse subset of the state space. Because these violation regions are sparse, the theory of *almost Lyapunov functions* applies [31]: small violation regions may induce temporary overshoots (requiring shrinking the certified invariant set), but they do not invalidate the safety and stability assurances of the certificate. Strong empirical results on controller performance in Section 6 support this conclusion, though we admit that good empirical performance is not a substitute for guarantees based on rigorous verification, which we hope to revisit in future work.

## 6 Experiments

To evaluate the performance of our learned rCLBF-QP controller, we compare against min-max robust model predictive control (as described in [2, 32]) on a series of simulated benchmark problems representing safe control problems with increasing complexity. The first two concern trajectory tracking, where we wish to limit the tracking error despite uncertainty in the reference trajectory. The next two benchmarks are UAV stabilization problems that add additional safety constraints and increasingly nonlinear dynamics. The last three benchmarks involve highly non-convex safety constraints. The first four benchmarks provide a solid basis for comparison between our proposed method and robust MPC, while the last three demonstrate the power of our approach to generalize to maintain safety even in complex environments.

In each experiment, we vary model parameters randomly in $\Theta$, simulate the performance of the controller, and compute the rate of safety constraints violations and average error relative to the

goal $\|x - x_{\text{goal}}\|$ across simulations. These data are reported along with average evaluation time for each controller in Table 1. To examine the effect of control frequency on MPC performance, we include results for two different control periods $dt$ for all robust MPC experiments (we also report the horizon length $N$). In some cases we observed that the evaluation time for MPC exceeds the control period; in practice this would lead to the controller failing, but in our experiments we simply ran the simulation slower than real-time. Our robust MPC comparison supports only linear models with bounded additive disturbance; we linearize the systems about the goal point and select an additive disturbance to approximate the disturbance from uncertain model parameters. The following sections will present results from each benchmark separately, and more details are provided in the supplementary materials, including the dynamics and constraints used for each benchmark, as well as the hardware used for training and execution.

Table 1: Comparison of controller performance under parameter variation

| Task | Algorithm | Safety rate | $\|x - x_{\text{goal}}\|$ | Evaluation time (ms) |
|---|---|---|---|---|
| Car trajectory tracking[1] | rCLBF-QP | | **0.7523** | **10.4** |
| Kinematic model | Robust MPC ($dt = 0.1$ s, $N = 6$) | | 1.5148 | 194.6 |
| ($n = 5, \ell = 2, n_s = 2$) | Robust MPC ($dt = 0.25$ s, $N = 6$) | | 12.4438 | 172.8 |
| Car trajectory tracking[1] | rCLBF-QP | | 1.0340 | **9.6** |
| Sideslip model | Robust MPC ($dt = 0.1$ s, $N = 5$) | | **0.1560** | 336.5 |
| ($n = 7, \ell = 2, n_s = 2$) | Robust MPC ($dt = 0.25$ s, $N = 5$) | | 18.1939 | 316.9 |
| 3D Quadrotor | rCLBF-QP | 100% | 0.4647 | **9.7** |
| ($n = 9, \ell = 4, n_s = 2$) | Robust MPC ($dt = 0.10$ s, $N = 5$) | 100% | **0.0980** | 316.2 |
| | Robust MPC ($dt = 0.25$ s, $N = 5$) | 100% | 63.6303 | 291.0 |
| Neural Lander | rCLBF-QP | 100% | **0.1332** | **13.1** |
| ($n = 6, \ell = 3, n_s = 1$) | Robust MPC ($dt = 0.10$ s, $N = 5$) | 100% | 0.2086 | 247.2 |
| | Robust MPC ($dt = 0.25$ s, $N = 5$) | 100% | 0.3267 | 253.2 |
| Segway | rCLBF-QP | **100%** | **0.0447** | **4.4** |
| ($n = 4, \ell = 1, n_s = 4$) | Robust MPC ($dt = 0.10$ s, $N = 5$) | 21% | 1.3977 | 214.8 |
| | Robust MPC ($dt = 0.25$ s, $N = 5$) | 11% | 1.9725 | 239.1 |
| 2D Quadrotor[2] | rCLBF-QP | | **83%** | **18.6** |
| ($n = 6, \ell = 2, n_s = 4$) | Robust MPC ($dt = 0.10$ s, $N = 5$) | | 53% | 276.9 |
| | Robust MPC ($dt = 0.25$ s, $N = 5$) | | 0% | 265.2 |
| Satellite Rendezvous | rCLBF-QP | **100%** | **0.1369** | **8.2** |
| ($n = 4, \ell = 2$) | Robust MPC ($dt = 0.10$ s, $N = 5$) | 39% | 6.3751 | 187.3 |
| | Robust MPC ($dt = 0.25$ s, $N = 5$) | 15% | 9.0592 | 197.4 |

[1] For car trajectory tracking, we compute maximum tracking error over the trajectory.

[2] For 2D quadrotor, we compute % of trials reaching the goal with tolerance $\delta = 0.3$ without collision.

Note: We also implemented SOS optimization to search for a CLBF and controller, but bilinear optimization (as in [33]) did not converge with maximum polynomial degree 10 and a Taylor expansion of the nonlinear dynamics.

## 6.1 Car trajectory tracking

First, we consider the problem of tracking an *a priori* unknown trajectory using two different car models. In the first model (the kinematic model), the vehicle state is $[x_e, y_e, \delta, v_e, \psi_e]$, representing error relative to the reference trajectory ($\delta$ is the steering angle). The second model (the sideslip model) has state $[x_e, y_e, \delta, v_e, \psi_e, \dot{\psi}_e, \beta]$, where $\beta$ is the sideslip angle [34]. Both models have control inputs for the rate of change of $\delta$ and $v_e$. We assume that the reference trajectory is parameterized by an uncertain curvature: at any point the angular velocity of the reference point can vary on $[-1.5, 1.5]$. The goal point is zero error relative to the reference, and the safety constraint requires maintaining bounded tracking error.

The performance of our controller is shown in Fig. 2. We see that for both models, both our controller and robust MPC are able to track the reference trajectory. However, robust MPC was only successful when run at slower than real-time speeds (with a control period $dt = 0.1$ s roughly twice as fast as the average evaluation time). MPC became unstable when run at a slower control frequency $dt = 0.25$ s. In contrast, our rCLBF-QP controller runs in real-time with a control period of $\approx 10$ ms on a laptop computer. This significant improvement in speed is due primarily to the reduction in the size of (rCLBF-QP) relative to that of the QPs used by robust MPC. For example, for the sideslip model, our controller solves a QP with 2 variables and 2 constraints, whereas the robust MPC controller solves a QP with 35 variables and 23 constraints (after pre-compiling using YALMIP [32]). Because the learned rCLBF encodes long-term safety and stability constraints into local constraints on the rCLBF derivative, the rCLBF controller requires only a single-step horizon (as opposed to the receding horizon used by MPC).

By comparing performance between these two models, we can discern an important feature of our approach. Increasing the state dimension when moving between models does not substantially increase the evaluation time for our controller (as it does for robust MPC), but it does degrade the tracking performance, suggesting that the number of samples required to train the CLBF to any given level of performance increases with the size of the state space. These examples also highlight a potential drawback of our approach, which relies on a *parameter-invariant* robust CLBF. Because it attempts to find a common rCLBF for all possible parameter values, our controller exhibits some small steady-state error near the goal. This occurs because there is no single control input that renders the goal a fixed point for all possible parameter values and motivates our use of a goal-reaching tolerance in Definition 1.

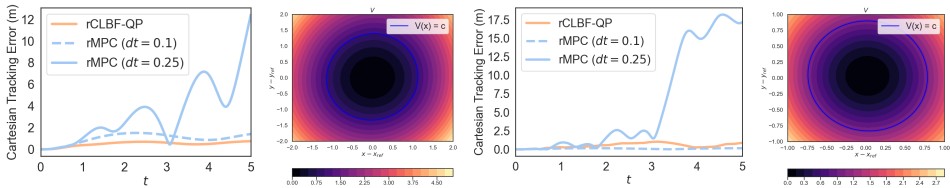

Figure 2: Trajectory tracking on kinematic (left) and sideslip (right) vehicle models, with contour plots of $V$. Blue shows the $c$-level set.

## 6.2 UAV stabilization

The next two examples involve stabilizing a quadrotor near the ground while maintaining a minimum altitude. Relative to the previous examples, these benchmarks increase the complexity of the state constraints, and we consider two models with increasingly challenging dynamics. The first model (referred to as the "3D quadrotor") has 9 state dimensions for position, velocity, and orientation, with control inputs for the net thrust and angular velocities [9]. The second model (the "neural lander") has lower state dimension, including only translation and velocity, with linear acceleration as an input, but its dynamics include a neural network trained to approximate the aerodynamic ground effect, which is particularly relevant to this safe hovering task [35]. The mass of both models is uncertain, but assumed to lie on $[1.0, 1.5]$ for the 3D quadrotor and $[1.47, 2.0]$ for the neural lander.

Fig. 3 shows simulation results on these two models. The trend from the previous benchmarks continues: our controller maintains safety while reducing evaluation time by a factor of 10 relative to MPC. Moreover, while the robust MPC method can achieve low error relative to the goal for the the 3D quadrotor model, the nonlinear ground effect term prevents MPC from driving the neural lander to the goal. In contrast, the rCLBF-QP method can consider the full nonlinear dynamics of the system, including the learned ground effect, and achieves a much lower error relative to the goal.

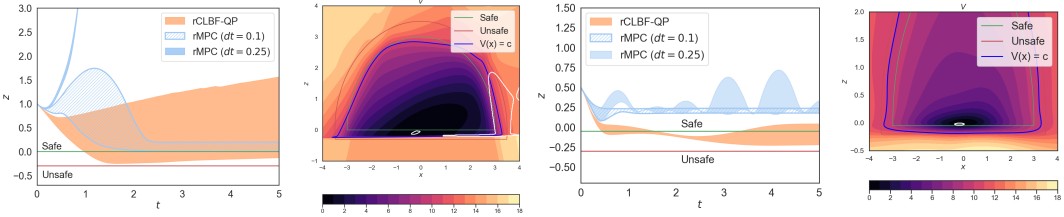

Figure 3: Controller performance for the 3D quadrotor (left) and neural lander (right), with contour plots of $V$. Blue shows the $c$-level set, white shows regions where condition (3) is violated.

## 6.3 Navigation with non-convex safety constraints

The preceding benchmarks all include convex safety constraints that can be easily encoded in a linear robust MPC scheme. Our next set of examples demonstrate the ability of our approach to generalize to complex environments. These problems are commonly solved by combining planning and robust tracking control, so in our comparisons we use robust MPC to track a safe reference path through each environment. In contrast, our rCLBF-QP controller is not provided with a reference path and instead synthesizes a safe controller using only the model dynamics and (non-convex) safety constraints, which is a more challenging problem than the tracking problem as in Section 6.1. The three

navigation problems we consider are: (a) controlling a Segway to duck under an obstacle to reach a goal [36], (b) navigating a 2D quadrotor model around obstacles [9], and (c) completing a satellite rendezvous that requires approaching the target satellite from a specific direction [37]. For (a) and (c), we conducted additional comparisons with a Hamilton-Jacobi-based controller (HJ, [38]) and policy trained via constrained policy optimization reinforcement learning (CPO, [39]). Simulated trajectories are shown in Fig. 4. Note that in the Segway and satellite examples, robust MPC fails to track the reference path, while the rCLBF controller successfully navigates the environment. HJ preserves safety in the satellite example but fails to reach the goal (which is positioned near the border of the unsafe region), while HJ controller synthesis failed in the Segway example (the backwards reachable set did not reach the start location with a 5 s horizon). Note that the HJ satellite controller requires different initial conditions, since it will fail if started outside of the safe region. The policy trained using CPO navigates to the goal in the satellite example, but it is not safe. In the Segway example, CPO does not learn a stable controller (details are given in the appendix).

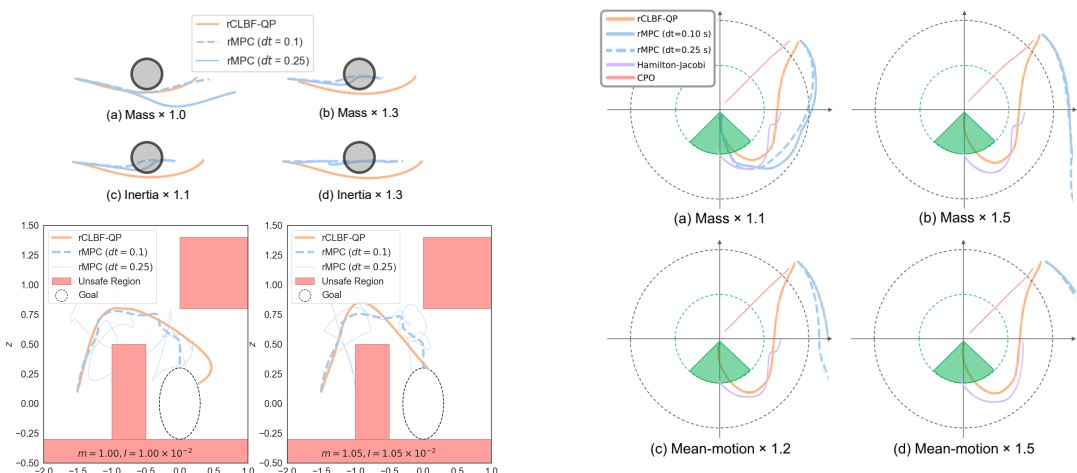

Figure 4: Navigation problems solved using our rCLBF-QP controller, compared with robust MPC. Clockwise from right: satellite rendezvous, planar quadrotor, and Segway.

## 7 Discussion & Conclusion

These results demonstrate two clear trends. First, the performance of our controller (in terms of both safety rate and error relative to the goal) is comparable to that of MPC when the MPC controller is stable. In some cases, our method achieves lower steady-state error due to its ability to consider highly nonlinear dynamics, as in the neural lander example. In other cases, the dynamics are well-approximated by the linearization and robust MPC achieves better steady-state error, but our approach still achieves a comparable safety rate. Second, we observe that the performance of the robust MPC algorithm is highly sensitive to the control frequency, and these controllers are only stable at control frequencies that cannot run in real-time on a laptop computer. This highlights one benefit of our method over traditional MPC, which trades increased offline computation for an order of magnitude reduction in evaluation time. In all cases, we find that our proposed algorithm finds a controller that satisfies the safety constraints despite variation in model parameters, validating our claim of presenting a framework for robust safe controller synthesis.

In summary, we present a novel, learning-based approach to synthesizing robust nonlinear feedback controllers. Our approach is guided by a robust extension to the theory of control Lyapunov barrier functions that explicitly accounts for uncertainty in model parameters. Through experiments in simulation, we successfully demonstrate the performance of our approach on a range of challenging safe control problems. A number of interesting open questions remain, including scalable verification strategies for $V$, the sample complexity of this learning method, and the relative convergence rates of $V$, $\pi_{NN}$, and the QP controller derived from $V$, which we hope to revisit in future work. We also plan on exploring application to hardware systems, including considerations of delay and state estimation uncertainty.

## Acknowledgments

The NASA University Leadership Initiative (grant #80NSSC20M0163) and Defense Science and Technology Agency in Singapore provided funds to assist the authors with their research, but this article solely reflects the opinions and conclusions of its authors and not any NASA entity, DSTA Singapore, or the Singapore Government. C. Dawson is supported by the NSF Graduate Research Fellowship under Grant No. 1745302.

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
