# OpenReview forum: "Safe Nonlinear Control Using Robust Neural Lyapunov-Barrier Functions"
_robot-learning.org/CoRL/2021/Conference — CoRL2021 Poster_

### Official Review · Reviewer_mZgd · 2021-07-23

**Originality:** Good
**Technical Quality:** Very Good
**Clarity Of Presentation:** Very Good
**Impact:** 4

**Recommendation:**

Weak Accept: I recommend accepting the paper, but will not argue for my recommendation if the majority of other reviewers have a different opinion.

**Summary:**

The main idea of the paper is to design feedback control systems with safety and stability guarantees by fitting robust Lyapunov/Barrier functions and policies, and then applying a standard pointwise min-norm control. The resulting controller is compared in simulation to robust linear MPC on a number of interesting examples.


**Issues:**

As above, I think the paper would be significantly improved if:
- The technical requirements of affine parameterisation and parameter-independent Lyapunov functions were discussed more clearly
- The question of sampling training points was clarified and explored in more detail
- It was clarified how the learned CLBF could be rigorously verified.

**Reviewer Expertise:**

Very good: Comprehensive knowledge of the area

**Strengths And Weaknesses:**

Strengths:
1) The authors are clearly well-versed in control theory. Material in Sections 1-4 provides a well-written overview of the main ideas around control Lyapunov functins (CLFs) and their close cousins the control barrier functions (CBFs).
2) The main technical novelty is a proposed method for computing CLFs, CBFs, and policies as a supervised-learning problem, given in Section 5, although this builds fairly naturally on the contributions of citation [8] in the paper.
3) The simulation results are interesting and demonstrate clear potential for the method. Experiments would be nice of course, but in the year of COVID it is understood that this was extremely difficult.

Weaknesses:
1) This is minor, but I think it could be more clearly stated in Definition 3 or Theorem 2 that the dynamics must be affine in \theta for this approach to work. This is briefly mentioned at the start of Section 3 and is dismissed as not restrictive, however in many cases of interest in robotics, in order to achieve an affine parameterisation of uncertainty it is necessary to use quite conservative overparameterisations.
2) Similarly, for parameter-dependent problems it is known that have parameter *independent* Lyapunov functions can be very conservative, however this is required for the proposed approach. The conservatism of this is not discussed.
3) I think the main contribution of the paper is really the sampling and fitting scheme in Section 5, but the discussion of sampling is not very deep, it just states that N_train points are sampled uniformly at random from X.
- Is X bounded? If not, how does one sample uniformly? If it is, how do we know X is robustly forward-invariant? Usually a Lyapunov function would be used to verify such a property, but this is assumed before the search for a Lyapunov function begins.
- Why uniform sampling? The method proposed is analogous to an actor-critic algorithm in RL with the "critic" in this case being a CLBF, and the Lyapunov condition replacing the Bellman equation. In RL uniform sampling of the state space is rare, and sampling design is indeed fundamental to the problem of "exploration vs exploitation", as well as classical experimental design.
4) I'm a little uneasy about claiming safety based on condition (3a) being satisfied only on a finite collection of sample points. The paper says "once trained, V can be verified using..." and gives some references. But these are quite general in nature, it is not entirely clear how they would be adapted to the present problem. Would they prove that (3a) holds globally?

**Summary Of Recommendation:**

Overall I enjoyed reading the paper and think it has the beginnings of a very useful approach. I have some reservations about points of detail, as discussed above, at least if the objective is for feedback systems that are really provably safe.

---

> ### Author Response · Authors · 2021-08-23
> **Author Response to Reviewer mZgd**
>
> Thank you for your feedback! We’ve revised our paper based on your comments, with major changes highlighted in blue. We’d also like to answer some of your questions directly.
>
> ### Affine-parameterization
> We have revised Definition 3 (page 4) to include the assumption of an affine parameterization. We would also like to note that our approach can also be applied in some cases where the dynamics are not affine in some parameters so long as the dynamics can be re-parameterized; for example, mass commonly enters as $1/m$, but simply treating $1/m$ as the uncertain parameter in the range $[1/(m_{max}),\ 1/(m_{min})]$ results in a suitable affine parameterization.
>
> ### Conservatism of Parameter-independent Lyapunov function
>
> You are correct that our use of a parameter-independent CLBF introduces some bias into our control policy. The most obvious manifestation of the effects of the parameter-independent CLBF are in Section 6.2, where we attempt to track a trajectory with unknown curvature. Since this curvature is unknown, it is impossible for the controller to maintain zero tracking error (in effect, there is no single control input that renders the goal a fixed point for all possible parameter values). This steady-state error also manifests in Figure 4, and is the reason why our problem statement (Definition 1) includes some tolerance around the goal. We have added some discussion to Section 6.2 covering this issue (page 7).
>
> ### Sampling Strategy
> Our choice of uniform sampling was driven by a few considerations. First, we need to ensure that points are sampled from both the safe and unsafe regions to ensure that the CLBF accurately separates these two regions (this means we cannot rely fully on a simulation-guided sampling strategy, since we need samples from the unsafe region). Second, we experimented with a hybrid sampling approach where we combined uniformly sampled points with points sampled along trajectories of the partially-trained controller, but we found that this approach did not significantly improve performance.
>
> One strategy that we found can be helpful in some situations is to sample a subset of the training data uniformly from the region around the goal, another subset uniformly from the unsafe region, another from the safe region, and then sample the rest of the points uniformly from the state space at large. For example, first sampling 10% of the training points uniformly near the goal, then 10% uniformly from the unsafe region, then 10% uniformly from the safe region, and the remaining 70% uniformly from the entire state space. We have added some discussion of this sampling strategy in the implementation section of our supplementary materials.
>
> We were able to sample uniformly from the state space by bounding the state space in each of our examples, and then sampling from the bounded space. To ensure that the controller did not leave the bounded region, we added safety constraints specifying that the system remain inside the training region (commonly in the form of a constraint limiting the L2 norm of the state). The full list of safety constraints for each example is given in the supplementary material, and we have added an explicit discussion of this “remain-in-distribution” safety constraint in the section of the supplementary materials covering training implementation (page 12-13).
>
> ### Verification
> We do not attempt to comprehensively verify our learned CLBF certificates in this paper, since scalable verification of neural network controllers for nonlinear systems remains an open research problem. However, we do present some preliminary results in this direction in our supplementary materials. Figures 6-12 include high-resolution plots of 2D slices of the state space for each system. For each slice, we solve the rCLBF-QP, with constraints in equation (4), at each point on a dense grid and plot the maximum violation of the rCLBF-QP constraints. For some systems, we found that the rCLBF-QP conditions hold everywhere except the region around the goal (reflecting the stead-state error we discuss above), while in other systems we found that the rCLBF-QP conditions hold everywhere except a sparse region of points where the violation is small (e.g. Fig. 8, where the maximum violation is on the order of 10^-4). To extend this approach to a comprehensive verification method, we would need to verify that equation (4) holds everywhere in the bounded state space, which (together with a safety constraint preventing the system from leaving the bounded space) would prove safety. This could be done with neural network reachability or output-optimization methods, some of which we reference in our discussion of verification strategies.

---

> > ### Comment · Reviewer_mZgd · 2021-08-27
> > **Response**
> >
> > I thank the authors for detailed response to my comments. All the answers are reasonable and make sense, and I think it is a nice paper. I do still have some concerns about talking about "safety and stability guarantees" when both the sampling strategy and verification are somewhat ad-hoc and non-rigorous, but as long as these issues are acknowledged in the paper I think it is ok.

---

> > > ### Author Response · Authors · 2021-08-27
> > > **Response**
> > >
> > > Thank you for pointing out this issue; we agree that these are challenging issues that our approach does not fully address. We have added a paragraph on page 5 of our paper that acknowledges these issues and points the reader towards some theoretical tools that can help close the gap. The paragraph is quoted below for convenience.
> > >
> > > > It is important to note that this training strategy encourages satisfying (3a) only on the finite set of training points sampled uniformly from the state space; there is no learning mechanism that enforces dense satisfaction of (3a). In the supplementary materials, we include plots of 2D sections of the state space showing that (3a) is satisfied at the majority of points, but there is a relatively small violation on a sparse  subset of the state space.   Because  these  violation  regions  are  sparse,  the  theory  of almost Lyapunov functions applies [31]:  small violation regions may induce temporary overshoots (requiring shrinking the certified invariant set), but they do not invalidate the safety and stability assurances of the certificate. Strong empirical results on controller performance in Section 6 support this conclusion, though we admit that good empirical performance is not a substitute for guarantees based on rigorous verification, which we hope to revisit in future work.
> > >
> > > [31] S. Liu,  D. Liberzon,  and V. Zharnitsky.   Almost lyapunov functions for nonlinear systems. Automatica, 113:108758, 2020
> > >
> > > Intuitively, the theory of almost Lyapunov functions laid out in [31] provides support for the idea that a Lyapunov-style certificate can still provide meaningful guarantees even if the corresponding conditions are violated on a subset of the region of interest. Our motivation for including the $\epsilon$ term in the various components of our loss function is to ensure that conditions like (3a) are satisfied with some margin, and this margin should encourage constraint satisfaction in a region around the sampled training points (due to the continuity of the network and condition 3a). Thus, our training process encourages learning an rCLBF that satisfies these conditions on a large portion of the region of interest, and we lean (admittedly informally) on almost Lyapunov theory, combined with our empirical results, to cover the gap.
> > >
> > > None of this is to claim that our certificates are rigorously formally verified; you are right to point out this issue and we have highlighted this gap in our paper as both a shortcoming of our approach and an interesting area of future work. We simply hope to explain our reasoning process and provide some intuition for why our certificates yield good performance (in terms of safety and stability) even without formal verification.

---

### Official Review · Reviewer_JDXV · 2021-07-23

**Originality:** Good
**Technical Quality:** Very Good
**Clarity Of Presentation:** Very Good
**Impact:** 4

**Recommendation:**

Weak Accept: I recommend accepting the paper, but will not argue for my recommendation if the majority of other reviewers have a different opinion.

**Summary:**

This paper proposes a somewhat novel definition of a Control Lyapunov-Barrier function. Such a function is approximately synthesized by learning it as a neural network using samples drawn from a robot's state space as training data. The method is demonstrated successfully on a variety of examples of increasing complexity.

**Issues:**

Here I've listed notes that I took while reading, most of which are "issues" but some of which are also just comments about what I enjoyed when reading the paper :)

Abstract
- Are the safety and stability guarantees DURING training, or only AFTER training?
- Does the proposed method consider actuator limits? How does one ensure there is always a feasible safe control action available?

Introduction / Related Work
- Fig. 1 would benefit from a bit of organization (maybe create a grid?) and some additional subtitles or labels explaining what each problem is (e.g., drone, segway, car)
- The discussion of robust MPC is really helpful and clear
- It's a bit odd to have just one subsection in Sec. 1
- An additional challenge to note with CBFs and CLFs is that it's unclear if a feasible safe control is always available (i.e., is the QP always feasible?)

Preliminaries and Background
- Thank you for clearly stating the problem! So many papers fail to do this!
- The CLBF definition is nice, but sounds very non-convex and hard to satisfy when synthesizing $V$ -- I'm really curious to see how you do it!
- Minor typo: the subscript "CLBF" in the paragraph under (2) shouldn't be italicized
- Give that CLBFs are basically a type of potential field, how do you actually ensure reachability? That is, how do you ensure the controlled system doesn't get, e.g., "stuck" behind an obstacle when the gradients cancel out in (1e)?

Robust CLBF Certificates for Safe Control / Learning rCBLFs
- I've had lots of reviewers complain when I put core proofs in supplementary materials. I personally think this is fine, but just be prepared for other reviewers to complain ;)
- Is it really reasonable to consider a *convex hull* of parameters? The idea of convexity here needs some further explanation to guide reader intuition and justify this idea.
- Fig. 2 is... not really helpful at all. I mean, there's a state, a neural network or two, and some outputs.
- Ah, I love this idea that $\pi_{\mathrm{NN}}$ is used to prove that the feasible set of the QP is nonempty! Very curious to see how that works.
- How would this method ever sample enough points in a high-dimensional state space to enable making strong claims that $V$ satisfies (1) (besides the "easy" claim of satisfying (1b))?
- It might be better to use "split" rather than "subequations" for (5), since it's a single equation continued onto a second line.
- I prefer the convention of using "\cdot" instead of "\circle" to represent an arbitrary argument (e.g., in where the paper explains the ReLU notation)
- OK, so it seems sneaky that $\pi_{\mathrm{NN}}$ is supposed to ensure the QP is always feasible, but, at the end of the day, you have to relax the QP to make this possible.
- It seems terrible to have to verify $V$ after training it. Was this verification actually implemented? I guess I'll find out in the next section!

Experiments
- What does "steadily" increasing complexity mean? Just say "increasing complexity" and avoid empty words!
- I really appreciate that you also tried to implement a SOS method. They're really hard to use!
- What hardware did you use to train your neural networks? How long did training take? Did you apply a verification method afterwards to check that the Lyapunov conditions hold? It's really odd to write a neural network training paper and then include no training results or discussion!
- For the simpler models, it would also be good to compare against, e.g., a grid-based HJB PDE solver approach, since such methods would have similar online running rates as your proposed method.
- Fig. 5 is fun and helpful

Discussion / Conclusion
- Cool, the method works!
- There are huge open questions, though! I think the conclusion does a good job of stating these questions -- but they aren't very surprising. I'm a little sad this paper doesn't do a better job of trying to address these questions, since the underlying CLBF idea isn't incredibly new or surprising.

**Reviewer Expertise:**

Very good: Comprehensive knowledge of the area

**Strengths And Weaknesses:**

STRENGTHS
- The first three sections are really well-written and easy to follow, much better than many other papers I've read
- The core idea is clear and well-founded in controls theory; I love seeing this effort to merge controls and learning literature.

WEAKNESSES
- The proposed CLBF is not very surprising, and the paper doesn't really address the hardest parts of synthesizing such an object; I don't feel that the idea is novel enough on its own to warrant such a "light" discussion of the challenges (which are, to the paper's credit, clearly stated in the conclusion!)
- The experiments section is severely lacking in explanation of how exactly the CLBF is synthesized; since this is a paper about learning a CLBF, it should explain more about learning, not just about application.
- The proposed method could probably be compared against a couple other approaches, besides just the rather naive robust MPC in the current comparison. I'd recommend comparing against an HJB PDE method and, if possible, a GP-based safe RL method.


**Summary Of Recommendation:**

I personally don't see how this is especially different from learning a potential field or other generic barrier function style approach. Given that the core idea is the CBLF, it was surprising to me that, just using the paper, I had no clear idea of how I'd implement the proposed method. Perhaps that is just my own ignorance! But I think there are many clarifications that could be added about the proposed method to make it a much more convincing paper.

Update (I'm leaving my previous comments unedited). The authors have addressed my concerns both in their comments and in the paper. I think the paper is much stronger now, and demonstrates an interesting advance in RL. I also must admit that I didn't carefully read the supplementary material while originally reviewing the paper! So now I will switch from "reject" to "accept."

---

> ### Author Response · Authors · 2021-08-23
> **Author Response to Reviewer JDXV (1/4)**
>
> Thank you for your comments; we appreciate your feedback!
>
> We’d like to begin by providing some high-level clarification about our approach, and then address your specific concerns. We have also revised our paper and supplementary material (major changes highlighted in blue) based on your feedback, including adding additional results comparing with HJB and safe RL on a subset of our examples.
>
> Due to space constraints, we have to break up our response into a couple of comments, which follow below.
>
> ### High-level clarification
>
> You asked how the rCLBFs used in our paper differs from a potential field or generic barrier function.
>
> On a theoretical level, CLBFs are generalizations of artificial potential fields (both the attractive and repulsive terms, just as CBFs generalize the repulsive terms of a potential field (cf. [1] below, which we now reference on page 2). On a practical level, the main difference between an rCLBF (or CLBF generally) and a potential field is that a CLBF must by definition satisfy condition (1e) --- if this condition is indeed satisfied everywhere, then there will be no local minima where the system can get stuck (the main drawback  of traditional potential fields). In fact, there is some theoretical work suggesting that even if (1e) is violated by a small amount on a small enough subset of the state space, then the reachability guarantees from Lyapunov theory still apply (cf. [2] below).
>
> There are two main differences between our rCLBFs and generic barrier functions:
>
> First, the rCLBF guarantees safety and stability for a range of parameter values, while generic barrier functions are specific to one set of parameters. We assume that the parameters are unknown but lie within the convex hull of a known set of points (a common example from our experiments is a hyperrectangle). The intuition for choosing this representation is that the key constraint on a CLBF-based controller (equation 2) is convex in the parameters (subject to the assumption that the dynamics are affine in the parameters, as discussed in section 3). Because of this convexity, checking equation 2 at each corner point of a polytope guarantees that it will hold continuously on the interior. This inspires the QP constraint in equation 4, which checks this constraint at each corner point of the set of unknown parameters.
>
> Second, an rCLBF includes stability and safety guarantees in a single certificate, whereas a control barrier function guarantees safety only. A control barrier function can be combined with a standard control Lyapunov function, but the gradient from these two functions can conflict and cause the controller to get stuck. By combining safety and stability into a single certificate function, we are left with a single gradient, and we can train the rCLBF so that this gradient satisfies condition (1e).  Our definition of rCLBF inherits this property from similar functions in references [26] and [27] in our paper; our specific advance over these CLBFs is the introduction of robustness and the use of neural networks to synthesize these objects. We clarify this on page 3.
>
> [1] A. Singletary, K. Klingebiel, J. Bourne, A. Browning, P. Tokumaru, and A. Ames, “Comparative Analysis of Control Barrier Functions and Artificial Potential Fields for Obstacle Avoidance,” Oct. 2020.
> [2] S. Liu, D. Liberzon, V. Zharnitsky, “Almost Lyapunov functions for nonlinear systems,” Automatica 113, 2020, https://doi.org/10.1016/j.automatica.2019.108758

---

> > ### Author Response · Authors · 2021-08-23
> > **Author Response to Reviewer JDXV (3/4)**
> >
> > ### Learning Implementation Details
> >
> > We apologize for not providing a clearer picture of our implementation. Due to space constraints, our initial submission included most of the implementation details in the supplementary materials, including hyperparameter values, sampling and initialization strategies, and details about training-time relaxation of the CLBF. These details can be found in `supplementary_material/corl21_robust_certificates_supplementary_material.pdf` (starting on the first page, marked 12). In response to your feedback, we have shifted some of this material to the main paper (see page 5).

---

> > > ### Author Response · Authors · 2021-08-23
> > > **Author Response to Reviewer JDXV (4/4)**
> > >
> > > ### Verification and Safety Guarantees
> > >
> > > Thanks for your questions about verification and safety guarantees, which we are happy to clarify here:
> > >
> > > - We are proposing a nonlinear control synthesis method, so the safety guarantees apply after training; we do not require safety guarantees during training in simulation. Unlike an RL-based approach, we use a model-based supervised learning approach by sampling points from the state space and applying the loss in equation (5) pointwise.
> > >
> > > - We discuss a few potential verification strategies in the supplementary material, but scalable verification of neural network controllers for nonlinear systems remains an open problem. In this paper, we focus on the control synthesis problem and do not attempt to exhaustively verify the learned certificates, but we do provide sampling-based evidence in that direction. Specifically, figures 6-12 in the supplementary material solve the rCLBF-QP at each point on a dense grid in a 2D slice of the state space, plotting the maximum violation of the constraints in equation (4) at each point. In some cases (the kinematic car, Fig. 6, and neural lander, Fig. 9), we found that rCLBF-QP is feasible everywhere except for a small region around the origin; intuitively, this is because it is impossible to robustly stabilize this plant at a single point when the parameters are unknown. In all other case, we found that either the QP is feasible everywhere except for a sparse set of points where the violation is small, or the QP is feasible everywhere inside the safe region but infeasible in some parts of the unsafe region. We hope that these results will help build confidence in our approach, even if they do not represent a comprehensive verification result.
> > >
> > > - The relationship between $\pi_{NN}$ and the need to relax the CLBF-QP. The main purpose of $\pi_{NN}$ is to act analogously to the actor in an actor-critic RL architecture (the rCLBF acts as the critic). $\pi_{NN}$ then helps refine the rCLBF, shaping the rCLBF gradient to encourage feasibility of the rCLBF-QP. However, we observed that the learned rCLBF typically converges to a solution where the rCLBF-QP is feasible before $\pi_{NN}$ converges. To address this mismatch, we use the training strategy described in the supplementary materials, where we solve the *relaxed* rCLBF-QP at training time, and then use that relaxation to determine which regions of the state space require additional training. In regions where the relaxation is large, we continue to use $\pi_{NN}$ to train the rCLBF, but in regions where the relaxation is zero we do not need to train further (as long as the safe/unsafe region boundary conditions are satisfied). Once training is complete, the required relaxation for the rCLBF-QP is very small (see the previous point), but we leave the relaxation enabled with a cost penalty on the order of $10^3$ to $10^5$ to avoid runtime exceptions from our QP solver. We’ve added a discussion of this strategy on page 5.

---

> > > > ### Comment · Reviewer_JDXV · 2021-08-30
> > > > **Response**
> > > >
> > > > Thank you for the clarification! I feel like I understand what is going on much better now.
> > > >
> > > > Also, I realize in retrospect that it is definitely fine for the supplementary material to contain all the training details. But, I'm glad you moved some of the material to the main text -- it makes the paper feel less opaque!

---

> > ### Author Response · Authors · 2021-08-23
> > **Author Response to Reviewer JDXV (2/4)**
> >
> > Additional Comparisons to HJB and Safe RL
> > -----------------------------------------------------------
> >
> > We have implemented HJ and safe RL methods for the satellite and Segway examples. We chose these examples because they are a) low-dimensional and b) contain complex safe/unsafe set specifications. We did not apply HJ methods to our other examples since they are too high-dimensional (these four-dimensional examples have solution times of 3-4 hours, which would grow to >100 hours for higher dimensional systems). Results from these comparisons are included in the main paper (page 8 and figure 5). We discuss the conceptual differences between our approach and these methods here.
> >
> > At a theoretical level, the basic assumptions of Hamilton-Jacobi methods differ from ours: we assume bounded parametric uncertainty, while HJ methods assume an additive adversarial disturbance. As a result, HJ methods will be conservative compared to our methods, since modelling an uncertain mass parameter (for example) will require over-approximation with an additive disturbance in the HJ framework. In addition, the controllers derived from the HJ optimal value function are themselves very conservative; typically, HJ optimal controllers do not allow the system to approach the unsafe region, even when necessary to reach the goal (i.e. HJ optimal controllers enforce a monotonically increasing safety measure).
> >
> > We see these differences play out in our results. In the satellite example, the HJ controller maintains safety but fails to reach the goal, which is located near the boundary of the unsafe region (since the HJ optimal controller is not able to approach the unsafe region in order to reach the goal). In the Segway example, we were unable to synthesize a controller using HJ reachability methods, since the backwards reachable tube did not grow to include the start state with a 5 second horizon (which was sufficient for our other controllers to reach the goal). This is likely due to the conservatism induced by over-approximating parametric uncertainty with an additive term.
> >
> > For our comparison with safe RL methods, we implemented constrained policy optimization (CPO). We also considered a popular GP-based safe RL method [3]; however, this work defines safety as stability in the region of attraction, where we define safety in a more general reach-avoid sense. As a result, we cannot use the method in [3] on our examples, so we use CPO instead. Of course, constrained RL algorithms like CPO can guarantee only probabilistic constraint satisfaction, but these results still provide a useful point of comparison.
> >
> > In the satellite example, CPO learned a policy that reached the goal, but it did not learn to avoid the unsafe region. In the Segway example, CPO was able to learn to stabilize the Segway in the absence of any obstacles, but we were not able to learn a safe, stable policy in the presence of an obstacle.
> >
> > [3] Berkenkamp F, Turchetta M, Schoellig A P, et al. Safe model-based reinforcement learning with stability guarantees[J]. arXiv preprint arXiv:1705.08551, 2017.

---

> > > ### Comment · Reviewer_JDXV · 2021-08-30
> > > **Response**
> > >
> > > Awesome, thank you for adding these comparisons! I specifically was hoping that you would show that the HJ method is more conservative than yours :) this is really great!

---

> > > > ### Author Response · Authors · 2021-08-30
> > > > **Response**
> > > >
> > > > Great! We're happy that you're happy :)
> > > >
> > > > Thanks again for your comments throughout this process; it has definitely helped improve our paper!

---

> > ### Comment · Reviewer_JDXV · 2021-08-30
> > **Response**
> >
> > Thanks for the clarification about your proposed CLBF! This makes a lot more sense to me now, and I better understand the novelty.

---

> ### Author Response · Authors · 2021-08-26
> **Follow-up**
>
> If you have any further questions or would like to see additional changes to the paper, we'd be happy to address those before the response period closes. Please just let us know.

---

### Official Review · Reviewer_o7Dq · 2021-07-24

**Originality:** Very Good
**Technical Quality:** Excellent
**Clarity Of Presentation:** Excellent
**Impact:** 4

**Recommendation:**

Strong Accept: I recommend accepting the paper and will argue for my recommendation even if other reviewers hold a different opinion.

**Summary:**

The paper formulates a novel approach for robust model-based learning based on control lyapunov functions (CLF) and control barrier function (CBF) in a unified framework that allows to produce a  robust lyapunov-like function that certifies stability and safety for a system with unknown but bounded parameters. The construction of the Lyapunov-like function is automatically generated from a supervised learning framework. The authors show the performance of the controller compared against robust MPC formulations on 7 different plants. They show that the robust control lyapunov function based QP (rCLBF-QP) offers more reliability on the safety rate and it outperforms robust MPC in most of the cases.

**Issues:**

The paper might benefit with more discussion about the implementation details of the neural network, like the number of train points, or the parameters a1 - a3, for instance.

**Reviewer Expertise:**

Very good: Comprehensive knowledge of the area

**Strengths And Weaknesses:**

The paper has an incremental and very clear presentation of the rCLBF, expressing theorems that provide a sound formulation of the controller.
The idea of providing a lyapunov-like function that produces a certificate for stability and safety together provides a better framework for learning and optimization.
Moreover, the conditions of the theorems are used to construct a Loss function for the neural network approximator of the lyapunov-like function, leaving less design work to hand tuned parameters or functions.


**Summary Of Recommendation:**

The paper proposes a novel formulation of a unifying lyapunov-like function that provides a robust certificate for stability and safety. The formulation allows an automatic way to produce such functions via supervised learning.

---

> ### Author Response · Authors · 2021-08-23
> **Author Response to Reviewer o7Dq**
>
> Thank you for your feedback. The request for additional implementation details regarding training was a common theme in our reviews, and we have revised our paper to include additional details. In the original version of our paper, we included some training implementation details in the supplementary materials (marked as page 12 in `supplementary_material/corl21_robust_certificates_supplementary_material.pdf`). These details include the values of $a_1$-$a_3$, as well as sensible defaults for the safety level $c$ and rCLBF convergence rate $\lambda$. We have expanded this section (changes highlighted in blue on pages 12 and 13) to provide additional implementation details, and we have included additional details in the main body of our paper (page 5).

---

### Meta-Review · Area_Chair_Re9g · 2021-08-13

**Recommendation:** Accept (Poster)
**Confidence:** 4

**Metareview:**

This paper proposes a novel learning-based approach to synthesizing robust nonlinear feedback controllers. While the reviewers agree that this paper contains some interesting ideas, they also raise a number of important concerns. The authors should carefully address the reviewers' comments in their rebuttal.

UPDATE POST DISCUSSION PHASE: I would like to thank the authors for their comments and clarifications during the discussion phase. The reviewers agree that this paper provides a valuable contribution, and I concur with this assessment.

---

> ### Author Response · Authors · 2021-08-23
> **Author Response Summary**
>
> We appreciate the feedback from our reviewers; their comments have pushed us to improve our paper in a number of ways. We also appreciate that our reviewers compliment the clarity of our presentation and understand the technical significance of our approach: using neural networks to automatically synthesize robust controllers for a wide range of nonlinear dynamical systems.
>
> We have responded individually to each reviewer below, and we have submitted a revised version of our paper and supplementary materials, with major changes highlighted in blue. In particular, these changes include:
>
> **Responses to Reviewer mZgd**
>
> 1.) We have included a discussion of the conservatism introduced by a parameter-independent CLBF on page 7.
>
> 2.) We have included additional details on our sampling strategy on pages 12-13 of the supplementary materials, and highlighted our work towards verifying the learned rCLBFs in Figures 6-12 in the supplementary materials.
>
> **Responses to Reviewer JDXV**
>
> 1.) We have included additional experiments comparing our approach to Hamilton-Jacobi and constrained RL methods. These results are included on page 8, with additional details on pages 14 and 22 of the supplementary material.
>
> 2.) We have included additional details on the implementation of our learning approach on page 5 (also applies to comments from Reviewer o7Dq).

---

> > ### Author Response · Authors · 2021-08-30
> > **Summary**
> >
> > This has been a helpful and productive review period. The reviewers' comments have helped us improve our paper, prompting us to include additional training implementation details, additional discussion of verification strategies and guarantees, and a new set of experimental comparisons. Both Reviewer mZgd (originally weak accept) and Reviewer JDXV (originally weak reject) have responded positively to our changes and new experimental results. We'd like to close by thanking everyone (reviewers and the AC) for their time and effort throughout this review process; we think our paper is much better as a result of our conversations, and we look forward to hearing the chair's decision.

---

### Decision · Program_Chairs · 2021-09-13

**Decision:**

Accept (Poster)

**Comment:**

This paper proposes a novel learning-based approach to synthesizing robust nonlinear feedback controllers. While the reviewers agree that this paper contains some interesting ideas, they also raise a number of important concerns. The authors should carefully address the reviewers' comments in their rebuttal.

UPDATE POST DISCUSSION PHASE: I would like to thank the authors for their comments and clarifications during the discussion phase. The reviewers agree that this paper provides a valuable contribution, and I concur with this assessment.